# Epidemiology, Diagnostics, and Therapy of Oral Cancer—Update Review

**DOI:** 10.3390/cancers16183156

**Published:** 2024-09-14

**Authors:** Julia Kijowska, Julia Grzegorczyk, Katarzyna Gliwa, Aleksandra Jędras, Monika Sitarz

**Affiliations:** Department of Conservative Dentistry with Endodontics, Medical University of Lublin, ul. Chodźki 6, 20-093 Lublin, Poland; julia.kijowska@interia.pl (J.K.); gjulia02@gmail.com (J.G.); katmgliwa@gmail.com (K.G.); jedrasaleksandra@gmail.com (A.J.)

**Keywords:** oral cancer, classification, epidemiology, diagnosis, treatment

## Abstract

**Simple Summary:**

Cancers of the lip and oral cavity are common cancers worldwide. Up to 46% of oral cancers are preventable if risk factors are avoided and if precancerous lesions are detected at an early stage. To reduce the incidence of oral cancer and its mortality rate, there is ongoing research studying new diagnostic and treatment methods. This review aims to present a novel glance at oral cancer—its current classification and epidemiology—and will provide new insights into the development of new diagnostic methods and therapies of oral cancer.

**Abstract:**

Oral cavity and lip cancers are the 16th most common cancer in the world. It is widely known that a lack of public knowledge about precancerous lesions, oral cancer symptoms, and risk factors leads to diagnostic delay and therefore a lower survival rate. Risk factors, which include drinking alcohol, smoking, HPV infection, a pro-inflammatory factor-rich diet, and poor oral hygiene, must be known and avoided by the general population. Regular clinical oral examinations should be enriched in an oral cancer search protocol for the most common symptoms, which are summarized in this review. Moreover, new diagnostic methods, some of which are already available (vital tissue staining, optical imaging, oral cytology, salivary biomarkers, artificial intelligence, colposcopy, and spectroscopy), and newly researched techniques increase the likelihood of stopping the pathological process at a precancerous stage. Well-established oral cancer treatments (surgery, radiotherapy, chemotherapy, and immunotherapy) are continuously being developed using novel technologies, increasing their success rate. Additionally, new techniques are being researched. This review presents a novel glance at oral cancer—its current classification and epidemiology—and will provide new insights into the development of new diagnostic methods and therapies.

## 1. Introduction

Cancers of lip and oral cavity are the 16th most common cancer in the world [1]. The incidence of head and neck cancer is constantly rising and is anticipated to rise by 30 percent by 2030 [2]. In order to prevent this, the well-known risk factors of oral cancer should be avoided. Furthermore, as discussed later in the text, the implementation of large-scale screening programs could prove advantageous for both patients and healthcare professionals, as it promotes the early detection of cancer, which is closely linked to improved curative treatment outcomes [3]. Regions that would see the greatest impact from these taken measures are countries in southern Asia, particularly India, and those in eastern Europe, as they have the highest number of incidences accompanied by the elevated mortality rates [1]. One example of such initiatives includes the cervical cancer screening programs implemented in India, which have achieved significant success [4,5,6]. For instance, in the Osmanabad district, a single round of HPV testing resulted in a reduction in advanced cervical cancer incidence and mortality by approximately 50% [4] The National HPV Vaccination Program in England is another example that, upon its introduction, achieved major success by lowering the prevalence of HPV16/18 forms and subsequently reducing the incidence of cervical cancer among young girls [7]. From the above-mentioned instances, it can be concluded that health programs conducted to prevent cancers are effective and necessary and hence should be included in the conversation regarding the reduction in other tumors such as oral lesions, in which they can have a great impact [8].

The aim of this paper is to provide a review of the current knowledge of oral cancer, including the classification, epidemiology, and recent advances to diagnostic and treatment techniques.

## 2. Overview of Oral Cancer

Cancers of the lip and oral cavity are a significant global health concern, with 389,485 new diagnoses and 188,230 deaths estimated in 2022 [1]. In addition, general awareness of the formation of this type of cancer and its risk factors in developing countries is low. A survey conducted in Beijing in the years 2018–2019 showed that about half of the respondents had never heard of this type of cancerous lesion, which should be alarming [9]. In developed countries, the situation is better, with 81% aware of oral cancer among the population [10]. It is a very important factor, as up to 46% of oral cancer cases are preventable [11]. An essential preventive measure is to also spread awareness about risk factors such as poor oral hygiene, smoking cigarettes, and low-nutritional diet [12]. Data show that annual medical examinations and early diagnosis of malignant lesions provide up to 90% chances of survival after treatment, stressing the importance of raising public awareness [13]. A study in Brazil involving 505 oral cancer patients highlighted a significant difference in outcomes: The five-year survival rate was 74.0% for those diagnosed at stage 1 or 2 but dropped to just 36.2% for patients at stage 3 or 4 [14]. Therapy commonly involves complete surgical removal of the tumor [15,16,17]. These and various other treatments are discussed later in the text. Even though the 5-year survival rate is increasing, it remains between 50% and 60%, which is still low compared to other types of cancer and indicates the need of improvement in the areas of spreading awareness, enhancing diagnostics, and creating new effective forms of medical therapy [18]. In addition, the introduction of screening programs on a larger scale could be a good initiative, both for patients and medical professionals, as knowing this condition well increases the chances of establishing effective treatment. While the examination of the oral mucosa is visually and tactually straightforward, and precancerous lesions are identifiable, the absence of structured initiatives for screening and secondary prevention of oral cavity cancers prevails in many regions globally. This gap in organized programs can be attributed to various knowledge deficiencies concerning the natural progression of oral precancer and the clinical handling of individuals with precancerous conditions [3].

One of the most common lip cancers is *Squamos cell carcinoma*, with its typical location on the lower lip, between midline and mouth corner (Figure 1) [19].

### Oral Potentially Malignant Disorders (OPMD)

Compared to healthy oral mucosal tissue, potentially malignant oral disorders show a higher risk of cancer transformation, especially into squamous cell carcinoma [20]. Patients diagnosed with OPMDs exhibit an elevated probability of developing oral cancer at any site within the oral cavity over the course of their lifetime. Although the majority of these OPMDs may not progress to carcinoma, they represent a field of abnormal cellular changes, within which the likelihood of cancer development is higher compared to clinically normal mucosa and significantly greater than in individuals without these disorders [21] The most common observed lesions can be divided into leukoplakias (OL), oral leukoerythroplakias, oral erythroplakias, and actinic keratosis [22]. Over the years, many definitions have emerged to define the condition known as leukoplakia [23]. In 2007, Warnakulasuriya et al. proposed that any white patch on the mucosa that cannot be characterized as any other abnormality should be described as such a lesion [24]. Based on the clinical appearance, there are three types of oral leukoplakia: homogeneous, non-homogenous, and verrucous, considering the fact that the first one occurs most often [25]. The formation of these lesions can be linked to such risk factors as smoking tobacco or drinking alcohol [26]. The OL prevalence in the general population is approximately 2%. This figure tends to rise in higher proportions among older age demographics [27]. In most cases, these lesions do not cause symptoms. In the remaining situations, patients experience rough spots on the mucosa and sometimes some burning sensations [28]. Proliferative verrucous leukoplakia stands out as a specific type of oral lesion with an unusually high risk of developing an oral malignant neoplasia [29,30]. It was confirmed that patients with multiple oral cancers are more likely to be diagnosed with new malignments in a shorter period of time than previously healthy patients or those with a single cancer [30]. To assess the risk of malignant transformation in oral precancerous lesions, an evaluation of the degree of dysplasia can be used. More than half of the potentially malignant lesions show no signs of abnormality at the level of the structural and cytological characteristics of the epithelium [25,31]. Individuals falling into this group may undergo a comprehensive monitoring initiative administered by dentists as part of their regular check-up routine [32]. Lesions occurring on the tongue and floor of the mouth should merit special attention and thorough diagnostics due to the fact that a large number of neoplastic transformations occur in these locations. This can also be linked to the greater percentage of OPMD with a high degree of dysplasia, such as a moderate or severe stage, at these sites [33]. Erythroplakia is acknowledged as a highly critical oral premalignant condition, demonstrating a significant likelihood of severe dysplasia or carcinoma, with prevalence rates typically falling within the range of 80% to 90% [34]. Fortunately, this type of lesion occurs very rarely [22,35].

## 3. Classification

There are a number of classifications that can be used to divide cancers that are localized in the oral cavity into groups. In 2022, WHO proposed dividing malignant lesions into oral cavity and mobile tongue tumors and a second section that contains oropharyngeal tumors located at the base of the tongue, tonsils, and adenoids. Table 1 presents more detailed data on the different types of lesions included in these two above-mentioned categories [19]. The most frequently observed group is epithelial tumors, especially squamous cell carcinoma [36,37]. Another classification that can be distinguished is based on the division of the oral cavity into seven subsites—lip, tongue, floor of the mouth, buccal, hard palate, alveolar retromolar trigone, and soft palate [38]. According to analysis of the Surveillance, Epidemiology, and End Results (SEER) database, published in 2019, cancerous malignments with the worst prognosis are localized on the tongue [39]. The classification of oral cancers is essential for the effective management of the disease, as it enables accurate diagnosis, tailored treatment, and prognostication. By categorizing oral cancers such as squamous cell carcinoma and salivary gland tumors, clinicians can better understand the distinct biological behaviors and clinical outcomes associated with each type. This stratification informs the selection of appropriate therapeutic interventions and supports a multidisciplinary approach to care. Additionally, classification is vital for advancing research into the molecular mechanisms of oral cancers, facilitating the identification of new biomarkers and treatment targets. It also ensures consistent communication among healthcare providers and enhances patient understanding, contributing to more personalized and effective care.

## 4. Epidemiology

The difficulty in systematizing data on the epidemiology of oral cancers is that they belong to the group of head and neck cancers (HNC), which means that a large amount of research is focused not only on oral cancer but also on that of the nasopharynx, oropharynx, salivary glands, tonsils, and others. Therefore, the information in the following review is a compilation of data from articles examining the epidemiology of both oral and head and neck cancers. 

Lip and oral cavity cancers rank as the 16th most common type of cancer globally [1]. The incidence of HNC is constantly rising and is expected to increase by 30 percent by 2030 [2]. Figure 2 generated by the GLOBOCAN website shows global trends in the incidence rate of lip and oral cavity cancer [1]. The regions with the highest incidence were Melanesia (ASR = 19.8), followed by South-Central Asia (13.5), Central and Eastern Europe (10.3), Western Europe (10.0), and Australia and New Zealand (10.0). The occurrence of lip, oral cavity, and pharyngeal cancer rises with age, reaching its peak in the 70–85+ age group. The growing number of cases in European countries, particularly in developed nations, could be attributed to improved early cancer detection and increased exposure to risk factors. It has been observed that cancer incidence and mortality rates tend to be higher in countries with a low and medium Human Development Index (HDI), as rapid socio-economic growth leads to the adoption of unhealthy lifestyles, behaviors, and environmental factors. A review of substance use across different regions found that tobacco smoking and alcohol consumption are major risk factors for oral cavity cancer in Europe. In Melanesia, South-Central Asia, and South-Eastern Asia, the widespread practice of betel quid chewing may contribute to the high prevalence of this cancer. Meanwhile, in parts of Oceania, particularly Australia and New Zealand, sun exposure has been identified as the most significant risk factor associated with the region [40]. A significant role in the increased incidence of head and neck cancer in European countries and USA has been attributed to oropharyngeal cancers linked to HPV infections [41]. The variability in incidence in highly developed countries is interesting. The incidence of oral cancer in men between 2005 and 2010 has been decreasing in countries such as France (−12.6%), Spain (−10.8%), and Hong Kong (−10.5%) wbuthile increasing in the U.K. (+18.8%), Japan (+21.3%), and Australia (+8.7%) [42]. The differing incidences of oral cancer observed in different countries can be attributed to a number of factors, including variations in health policies, lifestyle changes, and other socio-economic influences. In order to comprehend the reasons behind the contrasting trends in oral cancer incidence across different countries, it is essential to consider a range of factors, including the accessibility and quality of healthcare, lifestyle habits (such as alcohol consumption or smoking), and the economic and social conditions prevailing in the country. These elements can exert an influence on the occurrence and detection of oral cancer, which in turn gives rise to varying incidence trends.

## 5. Risk Factors 

### 5.1. Alcohol

Alcohol, especially in combination with tobacco, is considered one of the most important risk factors for oral cancer [1]. It causes not only oral cancer but also cancer of the larynx, pharynx, esophagus, breast, liver, and colorectum [43]. The most mutagenic part of the alcohol is acetaldehyde, which is the first product of ethanol metabolism [43]. Ethanol is metabolized via alcohol dehydrogenase (ADH) and via cytochrome P4502E1 (CYP2E1) to acetaldehyde (AA). Acetaldehyde is further metabolized via acetaldehyde dehydrogenase (ALDH) to untoxic acetate. AA causes DNA adducts, DNA repair inhibition, and DNA methylation and damages the antioxidative defense system (AODS). Additionally, the oxidation of ethanol by CYP2E1 produces reactive oxygen species (ROS), which the damaged antioxidative defense system (AODS) cannot effectively neutralize, leading to the formation of DNA adducts. CYP2E1 also converts various procarcinogens into their ultimate carcinogenic forms [44]. Ethanol metabolism occurs mainly in the liver; however, this process can begin in the oral cavity, which may cause accumulation of a mutagenic concentration of acetaldehyde in saliva. This happens due to the presence of bacteria and yeast in the normal microbiome of the oral cavity, and they play the main role in local acetaldehyde formation from alcohol drinks. Additionally, increased levels of ACH last as long as ethanol stays in the human body [45]. Mutagenic levels of ACH are estimated to be 40–100 µM. This can be achieved by drinking diluted vodka within 20 to 40 min [44]. Although the carcinogenic effect of alcohol has been scientifically proven, people’s awareness of the problem is relatively low. Figure 3 shows the percentage of people from different geographical areas aware that alcohol consumption is a risk factor for oral cancer. The survey data revealed significant differences in awareness of alcohol as a risk factor for oral cancer in different countries. The highest awareness was observed in India (64%) [46], followed by Yemen (58.9%) [47] and Italy (58.01%) [48], where more than half of respondents recognized alcohol as a risk factor. Germany [49] showed a moderate level of awareness, with 43% of respondents recognizing alcohol as a risk factor, as well as the United States of America (38%) [50], while the Australian state of Far North Queensland [51] had a lower level of awareness at 33%. The lowest levels of awareness were found in Denmark (15.4%) [52] and Sri Lanka (17%) [53], where only a small proportion of the population recognizes alcohol as a risk factor for oral cancer.

### 5.2. Smoking

Tobacco smoking is another risk factor that has a significant influence on oral cancer formation. Smokers are 7 to 10 times more likely to develop oral cancer than nonsmokers [54]. Besides sufficiently high awareness of the consequences, smoking cigarettes is still the most common addiction [55]. There are two factors in cigarettes that, when combined, increase the risk of oral cancer—nicotine and carcinogens [56]. Nicotine is not classified as a carcinogen by IARC, but it has a high addictive potential, which is why people still reach for cigarettes, exposing themselves to over 60 other carcinogens contained within them [56,57]. It is encouraging that the number of people using tobacco is decreasing, and according to WHO, in 2025 tobacco users will constitute 20.9% of the global population, i.e., 1.9% less than in 2020 (22.8%) [55]. On the other hand, a lot of people, mostly youth, have started to use substitutes like electronic cigarettes. For instance, research in Canada carried out by Cole et al. [58] revealed that the prevalence of e-cigarette use increased from 7.6% (2013–2014) to 25.7% (2018–2019). There are studies that show that e-cigarettes are less toxic compared to classic cigarettes [59,60], but there is currently growing evidence of a negative impact on human health due to, for instance, the presence of high levels of metals in vapor [61,62]. So far, there is no strong evidence that electronic cigarettes directly increase the risk of oral cancer [63], but e-liquids contain chemical compounds that are cytotoxic to the oral mucosa and other parts of the upper respiratory tract and can be destructive to DNA [64]. Therefore, electronic cigarettes can not be excluded from the group of mutagenic factors. 

### 5.3. HPV

Human papillomavirus (HPV) is the name of a group of more than 200 known viruses [65]. Symptoms of infection are usually mild, but there are a few high-risk types of HPV that can cause genital warts or cancer [65]. The percentage of diagnosed new cases of oral cavity cancers attributed to a HPV infection in 2018 was 2.1% [66]. The majority of detected cases were connected especially with HPV16 and HPV18 infection [67,68]. Additionally, HPV16 is the most persistent of all types, which results in prolonged exposure time (men 22 months; women 19 months) [69]. This directly increases the risk of oncogenic effects of the virus on human cells [70]. High-risk types 16 and 18 (HPV16/HPV18) are considered as one of the major risk factors in oropharyngeal cancers, but the specific role of HPV in oral cancer development is still unclear [71]. Some reviews on HPV and oral cancer suggest a strong causal link between HPV and oral squamous cell carcinoma (OSCC), the most common type of oral cancer, while others remain inconclusive [72].

### 5.4. Diet

Increased risk of oral cancer is also related to eating certain foods, mostly including those rich in pro-inflammatory factors. A pro-inflammatory diet causes prolonged inflammation, which may promote developing cancer in different parts of the body and also in the oral cavity [73]. Diets with high DII (dietary inflammatory index) increase levels of cytokines and other inflammatory biomarkers that intervene in the initiation and promotion of the cancer [74]. Thus, products that may contribute to the development of cancer include, for example, red and processed meat, refined grains, simple sugars, eggs, and high-fat dairy [75,76]. It has been proven that drinking very hot tea or eating spicy food also may increase risk of oral cancer [12]. On the other hand, there is a group of foods that may prevent oncogenesis. These include citrus fruits, yellow fruits and vegetables, blackberries, cranberries, products rich in omega 6 and 3 acids, garlic, curcumin, and many more [12,77,78,79,80].

### 5.5. Oral Hygiene

Inadequate oral hygiene leads to the accumulation of pathological plaque, of which bacteria are a major component [81]. The potential mechanisms by which the oral flora may lead to the development of cancer include (1) the metabolism of carcinogens (for example *Streptococcus* and *Neisseria*); (2) the production of carcinogens (for example, candida produces nitrosamines); (3) the induction of chronic inflammation (for example, bacteria that result in periodontal disease, *Prevotella intermedia*, etc.), in which the cytokines produced promote cell proliferation and inhibit cell apoptosis; (4) the direct influence of bacteria on cell cycle signals; and (5) bacteria directly damaging DNA by toxins. Table 2 shows examples of bacterial species inhabiting the oral cavity and their impact on oral cancer formation [82]. Given these potential mechanisms, it becomes crucial to consider the role of oral hygiene practices in mitigating cancer risk. Several studies have examined the relationship between toothbrushing frequency and the incidence of oral cancer [83]. The results show that tooth brushing is associated with a reduced risk of oral cancer. Moreover, with each additional daily brushing, the risk of oral cancer has been shown to decrease by 6% [83].

## 6. Diagnostics

It is widely known that diagnostic delay is related to lack of public knowledge about precancerous lesions, oral cancer symptoms, and risk factors. There is no doubt that suitable and early diagnosis is essential for successful therapy. The World Health Organization has indicated early detection as a fundamental effort to control the risk of oral cancer [84]. Unfortunately, it is quite difficult and requires a lot of knowledge, awareness, and experience from the examiner to diagnose this type of lesion during a standard clinical oral examination (COE) [85]. For this reason, most cases of oral cancer are detected in very late stages, which crucially contributes to low survival rates [86,87]. It is essential to conduct a comprehensive interview with the patient, particularly if they have a history of alcohol addiction, tobacco use, or HPV infection. Such individuals may necessitate more detailed and targeted diagnostic approaches. For cases related to alcohol, the focus is on frequent and thorough examination and biopsy of lesions. In contrast, for cases related to HPV, molecular testing and specific imaging studies may be prioritized. For instance, HPV-related cancers can spread to lymph nodes earlier, so imaging studies may be employed to assess regional lymphadenopathy more meticulously. A personalized approach that considers these risk factors enhances the effectiveness of early detection and treatment of oral cancer. Table 3 shows important symptoms of oral cancer, which should be looked out for during regular clinical examinations.

At present, there are several diagnostic techniques, which can be divided into currently available procedures and new developing technologies.

### 6.1. Vital Tissue Staining

Toluidine blue (TB) staining is an auxiliary, non-invasive technique commonly used during the COE. The dye utilized in this method stains cells that contain an increased amount of DNA or RNA, which is useful to identify lesions with possible malignant changes on the oral mucosa. This staining involves using a dye of 1% methylene blue, 1% malachite, 0.5% eosin, glycerol, and dimethyl sulfoxide. The more DNA there is in the cell, the more intensely the lesion will be stained [87]. This is related to one of the hypotheses about the mechanism of action of TB, which concerns the affinity of TB for nucleic acids. The commonly used technique is to combine TB with Lugol’s iodine. The use of Lugol’s solution causes iodine, contained in it, to combine with the glycogen presented in normal tissues and stains them to turn brown-black, while cancerous tissues remain unstained, which allows to define the malicious alteration. Due to certain limitations regarding the probability of false-negative results and the questionable specificity and sensitivity of this method, it was suggested to consider for biopsy every lesion that tests positively with TB stain [87,92].

### 6.2. Optical Imaging

Chemiluminescence was suggested to improve the identification of mucosal abnormalities compared with plain incandescent light. It is now known under the names ViziLite Plus and MicroLux DL [93]. This technique consists of using a 1% acetic acid solution to rinse the oral cavity for 1 min, which is followed by the examination of the mucosa under diffuse chemiluminescent blue-white light (wavelength of 490 to 510 nm). Acetic acid removes the glycoprotein barrier and slightly dries the oral mucosa; the abnormal cells of the mucosa then absorb and reflect the blue/white light differently than normal cells. Therefore, the normal mucosa appears blue, whereas abnormal areas of the mucosa reflect the light and appear whiter with brighter, sharper, and more defined margins [94]. Chemiluminescence has shown good sensitivity at detecting any oral PMDs, but the main concern is that it readily detects leukoplakia and often fails to spot erythematous patches. Moreover, the cost associated with chemiluminescence is quite high, which limits its utilization.

One of the devices used in oral cancer screening that has been recently marketed is the VELScope. Its technology is another non-invasive diagnostic option, involving using blue light in the wavelength of 400–600 nm. Normal oral mucosa fluoresces a bluish-green, while abnormal tissue shows reduced levels of autofluorescence and appears dark compared to healthy tissue [95]. It helps medical professionals spot the first signs of oral illness that might go undetected in bright light. Nonetheless, the VELScope instrument was unable to distinguish low-risk from high-risk lesions [96].

### 6.3. Oral Cytology

Oral cytology is a non-invasive procedure that can be a useful tool in the early detection and monitoring of oral health conditions. This diagnostic technique is often used to identify changes in oral mucosa that may indicate the presence of infections, pre-cancerous lesions, or other oral diseases. It involves collecting samples of cells from the mouth by a small brush, spatula or swab, which are subsequently transferred onto a glass slide and prepared for microscopic examination. While oral cytology is a valuable tool, it is important to note that it has its limitations, and definitive diagnosis may require additional tests or procedures, such as a biopsy. Although oral cytology is useful for initial screening, it is not as specific or sensitive as histopathological examination. There is a risk of false positives or false negatives, which necessitates further confirmatory testing [84].

Oral CDx is a trademarked name for a specific type of oral cytology test. The Oral CDx brush biopsy is a diagnostic tool used to aid in the evaluation of oral lesions and abnormalities. It was designed to provide additional information about oral lesions that may appear suspicious but do not show clear clinical signs of cancer or other diseases [94]. This is a minimally invasive procedure and is often performed during a regular dental examination. This technique involves using a small brush to collect cells from the suspicious oral lesion. The collected cells are transferred onto a glass slide, which is then sent to a laboratory for analysis [87]. The goal of the Oral CDx brush biopsy is to assist in identifying whether a suspicious oral lesion requires further investigation, such as a traditional biopsy, or if it can be monitored over time. It is important to note that while the Oral CDx brush biopsy can provide valuable information, it is not a substitute for other diagnostic procedures, and its results should be interpreted in conjunction with the patient’s clinical history and other relevant information [84,94].

Oral exfoliative cytology (OEC) is straightforward, non-invasive technique that is comfortable for patients and can facilitate the early detection of oral cancer. OEC focuses on the morphological and staining characteristics of individual cells, requiring the expertise of skilled cytopathologists [97]. However, despite its advantages, OEC is neither specific nor sensitive. Consequently, it is primarily used for screening large populations, regularly monitoring precancerous lesions, and selecting appropriate biopsy sites within extensive lesions. After oral cancer is identified through OEC, an incisional biopsy is performed, where a carefully selected tissue sample is taken for diagnosis. Although it does not encompass the entire lesion, the incisional biopsy is fairly accurate. This method is particularly useful when it is not feasible to remove the entire lesion, such as in the case of a large white patch or lichen planus, and is also preferred when the clinical diagnosis is uncertain [98].

Recent advancements in AI and digital pathology are enhancing the capabilities of oral cytology. AI algorithms can assist in analyzing cytological images, improving accuracy, and reducing the subjectivity associated with human interpretation. These technologies are making oral cytology a more reliable tool in the early detection and ongoing monitoring of oral cancer [99].

### 6.4. Salivary Biomarkers

Salivary biomarkers are substances or molecular entities present in saliva that can serve as indicators of various physiological or pathological conditions, including oral cancer. The study of salivary biomarkers in oral cancer diagnostics is an area of active research and may hold promise for non-invasive and early detection of oral cancer [100,101]. Among the studied salivary biomarkers, various types of substances, such as proteins, DNA, RNA, and exosomes, are being investigated [102]. Specific biomarkers, e.g., CA 125, CA 15-3, IL-8, p53, and microRNA, have undergone analysis to determine their diagnostic potential. These studies aim to develop simple, non-invasive tests for the rapid detection of oral cancer and monitoring disease progression. The advantage of salivary biomarkers lies in the ease and painlessness of sample collection. However, before the full implementation of salivary biomarkers in routine clinical practice, further research is necessary to confirm their effectiveness and reliability. Standardizing analysis methods, eliminating confounding factors, and validating results are crucial for the success of salivary biomarkers in the diagnosis of oral cancer [103]. In the perspective of the future, this innovative field may play a significant role in improving the effectiveness of detection and treatment of this type of cancer [102].

### 6.5. Artificial Intelligence

The role of artificial intelligence (AI) in oral cancer diagnosis is an area of active research and development. AI technologies have the potential to enhance various aspects of oral cancer diagnosis, including early detection and classification of lesions [104,105].

AI can be categorized into traditional machine learning (ML) and deep learning. Traditional ML employs algorithms and computational processes to analyze data and identify patterns within it, providing a quantified diagnostic outcome. ML techniques are typically classified as either supervised or unsupervised. Deep learning, also known as neural networks, involves the use of nonlinear processing units organized in multiple layers, allowing the system to learn from input data and associate the output with the corresponding input [106]. Currently, these techniques are being evaluated to improve diagnostic methods, particularly for disease screening in areas with limited availability of doctors and trained specialists. AI can be utilized in various ways to aid in the prevention and early detection of oral cancer.

The potential of AI extends to non-invasive methods, particularly in the analysis of salivary biomarkers by the identification of their subtle variations associated with oral cancer. Predictive analytics powered by AI enable risk stratification, predicting the probability of oral cancer development based on diverse patient-related factors [107]. One of the key roles of AI in oral cancer diagnosis is in image analysis and recognition. Alhazmi A et al. proved that machine learning technique has the potential to help in oral cancer screening and diagnosis by predicting the individuals’ risk of developing oral cancer based on data on risk factors, systematic medical condition, and clinical–pathological features [108].

Moreover, thanks to the ability to process vast amounts of histopathological data, AI assists pathologists in identifying mucosal abnormalities indicative of oral cancer, potentially leading to quicker and more accurate diagnoses [109]. This advancement offers promising potential not only for improving early detection and monitoring of oral lesions, but by reducing the likelihood of false positives and false negatives, AI can also enhance the accuracy of diagnoses [110].

As AI continues to evolve, collaborative efforts between healthcare professionals, researchers, and technology developers will be pivotal in harnessing its full potential for improved outcomes in oral cancer detection and management [111].

### 6.6. Colposcopy

While colposcopy is traditionally employed for examining the cervix and vaginal tissues, recent studies have brought attention to its potential application in the field of oral oncology. Given the similarities in anatomy and cancer types between the oral cavity and cervix, acetic acid appears to be a suitable clinical marker for detecting oral cancer as well.

The procedure involves the use of a colposcope, a specialized instrument equipped with magnifying lenses and a light source. The colposcopic examination procedure ia carried out by using various light filters, acetic acid, and Lugol’s solution. The colposcope employs green or blue filters to better visualize vascular changes and color variations, as unfiltered white or yellow light decreases the contrast between terminal vessels and the surrounding tissue. The green filter specifically eliminates red light, thereby enhancing the visibility of vascular structures by making blood vessels appear darker. Acetic acid and Lugol’s solution are applied to the surface to enhance the visibility of abnormal areas [112].

Colposcopy possesses the capability to emerge as a promising and non-invasive approach for detecting oral lesions. It requires both specialized equipment and training. The interpretation of colposcopic images can be subjective, and the diagnostic accuracy is highly dependent on the operator’s experience and expertise. The examiner looks for any abnormal color changes, patterns, or lesions that may indicate the presence of pathology. In the context of oral colposcopy, the goal is to identify signs of oral cancer or precancerous lesions. If suspicious areas are found, a biopsy may be performed for further laboratory analysis.

Direct oral microscopy has also detected subclinical lesions that exhibited no apparent color change but were subsequently confirmed to be dysplastic [113,114,115]. Since this procedure is relatively quick and non-invasive, it can be integrated into routine practice, particularly in high-risk populations or specialized oral cancer clinics. However, there is no single colposcopic parameter accurate enough for achieving a definitive diagnosis [116]. While colposcopy is a well-established technique for gynecological examinations, its adaptation for oral cancer diagnosis is an area of ongoing research, requiring verification through clinical studies [114].

### 6.7. Spectroscopy

Spectroscopy plays a significant role in oral cancer diagnostics, offering a non-invasive and rapid method for analysing tissues and detecting abnormalities. Importantly, spectroscopy contributes to early cancer detection by identifying pre-cancerous lesions and detecting cancers in their initial stages [117,118]. Moreover, it assists in surgical procedures by offering real-time assessment of surgical margins during tumor removal surgeries, ensuring the comprehensive removal of cancerous tissue. As technology advances and becomes more accessible, spectroscopy may play a crucial role in population-wide screening programs for oral cancer, further emphasizing its potential impact on improving early detection and treatment outcomes [117].

Each diagnostic method has its unique advantages and limitations, and a comprehensive approach often involves combining multiple techniques to ensure accurate detection, staging, and monitoring of oral cancer. A summary of these is shown in Table 4. Advances in technology and ongoing research contribute to the continuous improvement of these diagnostic methods, with the goal of enhancing early detection and improving patient outcomes. Despite the advent of numerous high-end diagnostic techniques, much needs to be answered before these diagnostics become a routine practice, thus aiding in designing the tailored therapy for oral precancer and cancer.

## 7. Treatment

Every patient diagnosed with oral cancer should be assessed in order to establish an appropriate, individualized and optimal management plan [38]. Various diagnostic techniques are used to localize and measure the tumor and its accessibility, thus allowing the classification according to TNM. Those techniques include diagnostic imaging such as computer tomography (CT), magnetic resonance (MR), and ultrasonography (USG). In cases of advanced tumors (stage III and IV), positron emission tomography (PET) may be helpful in making a diagnosis [119].

The main method for cancer stage classification is the TNM system, published by Union for International Cancer Control [120]. This system is used to describe the tumor, its size (T), possible metastases in nearby lymph nodes (N), as well as metastases to other parts of the body (M). By adding specific numbers (from 0–4) or “X” if unmeasurable, professionals can accurately describe the clinical stage of the tumor.

The treatment choice is influenced by the site of disease, stage, and pathologic findings [38,121]. Firstly, a multidisciplinary team of healthcare providers should determine whether curative or palliative treatment is offered. Curative treatment is chosen if the disease is surgically resectable and confined to the primary site and possible cervical nodes. Palliative treatment remains an option when the cancer has spread to distant sites or is surgically unresectable due to involvement of vital structures [38]. Surgery, radiation, and chemotherapy are the main modalities for oral cancer treatment. However, novel therapies are continuously being researched [15].

Single-modality treatment with surgery or radiotherapy is generally recommended for patients who present early-stage (stage I or II) disease. Combined modality treatment is advised for patients with locally or regionally advanced disease [121].

The current NCCN Clinical Practice Guidelines on treatment of cancer of the oral cavity are summarized below in Figure 4 [121].

### 7.1. Surgery

Surgery is the primary modality for oral cancer treatment [15,38,121]. The main goal of surgical treatment is the complete resection of the tumor with an adequate margin, analysis of which can be used as a prognostic factor for patients and influence further therapy [122,123]. The guidelines for diagnosis of margins issued by The Royal College of Pathologists in 1998 are widely accepted. These state that surgical margins of <1 mm are considered positive, 1–5 mm close, and >5 mm clear/adequate [124]. Over recent years, there has been much debate on updating these guidelines for oral squamous cell carcinoma, but this has not been implemented [122,125,126].

Currently, surgeons commonly assess the margins by visual inspection and palpation [122]. However, in order to minimize the occurrence of inadequate resections in the oral cavity, the status of the margin should be investigated [127,128] using intraoperative frozen section (most popular but prone to false-negative results) [129,130,131,132], fluorescence molecular imaging, or narrow band imaging during surgery. Although fluorescence imaging has been shown to be highly sensitive, narrow-band imaging can be preferable in early-stage tumors for practical and cost-related reasons [133,134].

Besides margin diagnosis, a factor that significantly influences survival in patients treated surgically for oral cancer is the approach taken [135]. Tirelli et al. retrospectively compared patients treated by more invasive, en-bloc surgery (primary tumor and cervical lymph nodes removed in continuity) and those treated with the less invasive, discontinuous approach that preserves the anatomic separation between the oral cavity and cervical spaces. They concluded that the more invasive approach does not correlate to better locoregional control or survival [135].

### 7.2. Radiotherapy

Adjuvant postoperative radiotherapy is recommended for patients with stage III or IV disease, inadequate margins, and invasion into perineural and lymphovascular spaces or bone [136,137]. A study carried out by Hosni et al. additionally suggested that when a surgical approach is not suitable, for example, due to medical inoperability, surgical unresectability, and attempted preservation of oral structure, radiation therapy has a meaningful rate of locoregional control and can be an alternative curative approach [138]. Patient-related factors such as comorbid conditions, advanced age, or patient refusal of surgery due to risk of postoperative facial deformities and difficulties in eating, swallowing, and speech can all steer the primary treatment toward radiotherapy [139,140]. Unfortunately, a retrospective cohort study suggests that initial radiation therapy is associated with a greater mortality than surgery [141]. Curative radiation therapy requires the administration of high doses of radiation to a small area of many critical structures. In order to achieve this, modern radiotherapy uses three-dimensional conformational radiotherapy and intensity-modulated radiotherapy (IMRT) [142]. Lapeyre et al. suggested that three-dimensional conformational radiotherapy should be abandoned [143]. IMRT improves oncologic outcomes and reduces the radiation-related toxicity, as it applies the radiation more precisely [136,142]. Single-phase irradiation, known as irradiation with a simultaneous integrated boost (IMRT-SIB), allows for a slightly accelerated treatment [143]. However, recent studies have shown that IMRT neither enhances disease control nor compromises it [136,144], but it can cause many side effects, thus causing questions about the use of this technique post surgery [144,145].

### 7.3. Chemotherapy

Chemotherapy can potentially be used as primary treatment when surgery is not possible as an induction chemotherapy before definitive treatment, an adjuvant postoperative therapy, or as a concurrent chemoradiotherapy [121].

Chemoradiation with cisplatin 100 mg/m^2^ given once every 3 weeks is a standard of care in locally advanced head and neck cancer. An effort has been made to reduce the dose and therefore toxicity, but findings show that although more toxic, it is superior in locoregional control compared to once-a-week 30 mg/m^2^ cisplatin and therefore is preferred [146]. Alternative agents include carboplatin and infusion 5-fluorouracil [121]. However, a recent meta-analysis has suggested that the frequency percentage of 5-fluorouracil resistance is as high as 40.2% [147]. The emergence of chemoresistance, where cancer cells switch between molecular pathways and mechanisms to ensure their proliferation, invasiveness, and resistance to chemotherapeutic agents, is the present challenge associated with chemotherapy [148,149]. Currently, the most common method of overcoming such resistance is the application of antitumor drugs along with chemotherapeutic agents. Finding appropriate compounds is the focus of researchers [150].

There is much debate regarding the effectiveness of chemotherapy in treating oral cancer, which is largely caused by difficulty in conducting reliable research. Retrospective study results are affected by confounding sample variability, as patients are treated appropriately for the stage and location of oral cancer that they are presenting.

Some authors suggest that adjuvant chemotherapy does not improve treatment outcomes in patients with oral cancer and should therefore not be used [151]. Similarly, the results of a recent study comparing the survival of patients for locally advanced oral cancer treated with surgery followed by postoperative radiotherapy or chemoradiotherapy implied that additional chemotherapy does not influence the survival of patients [152]. Tangthongkum et al. reported that there is no significant difference in survival outcome between concurrent chemoradiotherapy compared with primary surgery plus radiotherapy. Therefore, concurrent chemoradiotherapy should only be used as an alternative treatment for patients for whom surgery is relatively contraindicated [153].

On the other hand, it is crucial to note that adjuvant chemotherapy is recommended in the most recent NCCN Guidelines when adverse pathologic features (entranodal extension, positive margins, close margins, pT3 or pT4 primary, pN2 or pN3 nodal disease, nodal disease in levels IV or V, perineural invasion, or vascular or lymphatic invasion) are confirmed post surgery [121]. Foster et al. offered chemoradiation for oral cavity cancer patients over two decades and found that chemoradiation represents a viable and feasible strategy for organ preservation for patients with locally advanced oral cancer [154]. A significant portion of patients prefer this option, even when aware of the increased risk of osteoradionecrosis that comes with chemoradiotherapy [155] since the alternative (surgery) could entail a near-total or total glossectomy [156].

When surgery is not possible or not preferred by the patient, chemotherapy has a chance to play an important role. It has been suggested that certain OSCCs, which are technically unresectable due to extensive locoregional disease, might be made resectable using induction chemotherapy. Through such therapy, unacceptable amounts of cosmetic deformity and functional morbidity may be avoided [157,158]. In addition, chemotherapy can be used in palliative care for patients with recurrent disease to improve quality of life and prevent metastasis [159,160,161,162]. It has been demonstrated that metronomic adjuvant chemotherapy using tegafur–uracil reduces the distant metastasis rate and improves survival rates in patients with advanced oral cancer after curative treatment [163], thus showing a place for chemotherapy in treatment protocol of oral cancer.

### 7.4. Immunotherapy

Immunotherapy applies biotechnology and immunological methods to improve the specific immune response to the tumor [164,165]. In recent years, two programmed death-1 (PD-1) inhibitors, nivolumab and pembrolizumab, were approved for the treatment of patients with oral cancer displaying the following characteristics: recurrent or metastatic squamous cell carcinoma and disease progression within 6 months of platinum-containing chemotherapy [166,167]. This treatment has been associated with long-term remissions only in a small number of patients [168]. 

#### 7.4.1. Immunotherapy Strategies under Clinical Trials

The immunotherapy strategies currently being researched are shown in Table 5.

#### 7.4.2. Immunotherapy and Chemotherapy Combination

The tumor immune microenvironment can be categorized as “hot” or “cold” by observing the distribution of immune cells. An immunologically “hot” tumor has distributed immune cells, while a “cold” tumor lacks them [173]. Immune checkpoint inhibitors alone are most effective against “hot” tumors. “Cold” tumors require other therapies, such as chemotherapy, to recruit the immune cells to the tumor tissue (convert the tumor from “cold” to “hot”), making immunotherapy effective [174]. In line with this theory, new combinations of immuno- and chemotherapy (pembrolizumab + docetaxel and pembrolizumab + lenvatinib) have been tested and expressed promising positive effects with minimal side effects in oral cancer treatment [175,176].

### 7.5. Novel Treatment

Despite many options for oral cancer treatment, the prognosis has not been significantly improved in the last 50 years [177], calling for further research. The currently developing therapy modalities discussed below are a step toward less invasive and detrimental but more targeted treatment options for patients with oral cancer.

Photodynamic therapy has been suggested to be a promising, minimally invasive, safe treatment in oral squamous carcinoma patients with locally advanced sites [178]. This suggests a possible alternative to the surgical approach, which can often cause a fundamental decline in a patient’s quality of life due to its extremity [139]. In photodynamic therapy, the active photosensitizer reacts with the tissue oxygen, damaging the cells and causing tumor necrosis [179]. Recent research has concentrated on new methods of its application and use as a primary or adjuvant modality for oral cancer [180,181,182,183]. Phase II and III clinical trials on photodynamic therapy should be the next focus for research.

Recent studies show that phytochemicals may also have a role in limiting the progression of oral cancer [184]. Secondary plant metabolites such as phenolic compounds, terpenoids, and nitrogen- and sulfur-containing compounds have inherent bioactivity and are therefore readily available for cellular uptake. Interactions with cellular and molecular pathways involved in the development of oral cancer make them ideal for treating these tumors [185]. Additionally, phytochemicals less frequently express dose-limiting side effects, making them superior compared with conventional anticancer agents. Although many experimental studies on phytochemicals have been carried out, less than 10 unique phytochemicals have been evaluated in clinical trials [184].

The antimetastasis ability against various cancers of probiotics and postbiotics suggest that they may have a potential in having the same effect on oral cancer [186]. Unfortunately, the available research is limited to in vitro and animal studies [187,188,189,190,191].

Melatonin displays oncoprotective and oncostatic activities through many mechanisms. It has antioxidant properties and an ability to inhibit the pro-angiogenic vascular endothelial growth factor (VEGF), epidermal growth factor (EGF), and insulin growth factor-1 (IGF-1). This interferes with tumor cell proliferation and growth [192]. Melatonin is also an immunoenhancer [193] and may help in increasing chemo- and radiosensitivity [194]. The focus of current research is on the relevance of melatonin receptors in the oral cavity and on melatonin’s potential as a likely mediator of epigenetic effects on oral cancer cells [192,195].

RNAi gene therapies have not yet been tested in human trials, but they have been shown to reduce targeted proteins expression. The study limitations are related to delivery challenges, targeting, and side effects, which are currently being addressed to move the research forward [196].

It was recently shown that dysregulation of certain molecular signaling pathways can have a role in the etiology of oral cancer and can be related to chemo- and radiotherapy resistance [197]. This is because autophagy is suppressed [198]. Among the most important pathways related to over 90% of head and neck cancer is PI3K/AKT/mTOR signaling [199]. It is vital for regulating the cell cycle [200], so if disturbed, it leads to uncontrolled growth of cells. Targeting this pathway via its inhibitors may be an effective strategy of treating oral cancer patients and is already under clinical trials in combination with other treatment modalities to overcome therapeutical resistance [201,202,203,204]. Nuclear Factor Kappa B (NF-κB) is another component that plays an important role in the progression of oral cancer through regulating gene transcription, increasing the expression of pro-inflammatory cytokines, and promoting metastasis [205]. It was demonstrated that NF-κB is highly expressed in oral squamous cell carcinoma compared with normal tissue [205]. Compounds targeting this pathway are currently being developed [206]. Studies have also highlighted the contribution of Wnt signaling pathway activation toward oral cancer progression [207,208]. Unfortunately, little has been explored about possible therapies targeting this pathway in oral cancer mainly due to the lack of understanding about the mechanisms involved [209]. Other molecular pathways such as Notch, MAPK, Hippo, and Hedgehog are still in very early stages of investigation and require extensive further research to discover their role in oral cancer treatment [210,211,212,213].

The efficiency of currently used treatments for oral cancer is not constant, whereby responses vary from patient to patient. This is related to genetic variability [214]. The above-mentioned research on melatonin, RNAi gene therapies, and compounds targeting molecular pathways are all opportunities for personalized therapies, thus making oral cancer treatment more successful in terms of survival rate and healthy structure preservation.

## 8. Conclusions

Oral cancer is a serious problem mainly in middle- and low-income countries. A factor that has a significant impact on survival rate is the advancement of the cancer at the time of detection, which is why early diagnosis is important at the precancerous disease stage. Therefore, general dentists should carefully examine not only the teeth but also the oral mucosa during each visit. This approach increases the chance of quick implementation of appropriate treatment. The incidence of disease can also be reduced through preventive programs, the aim of which is to educate patients on limiting exposure to risk factors such as smoking or alcohol consumption. Oral cancer treatment should be carried out in cooperation with a multidisciplinary team that will select the appropriate treatment method. The choice of method depends on several factors, but modern, advanced technology increases the probability of every treatment success.

## Figures and Tables

**Figure 1 cancers-16-03156-f001:**
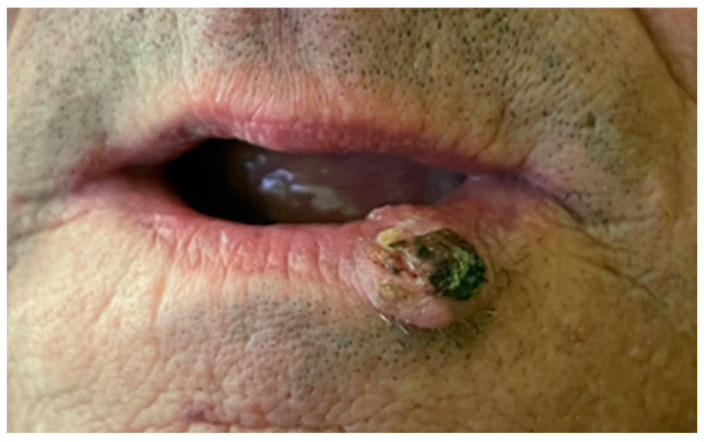
*Squamos cell carcinoma*, patient qualified for surgical removal of cancer.

**Figure 2 cancers-16-03156-f002:**
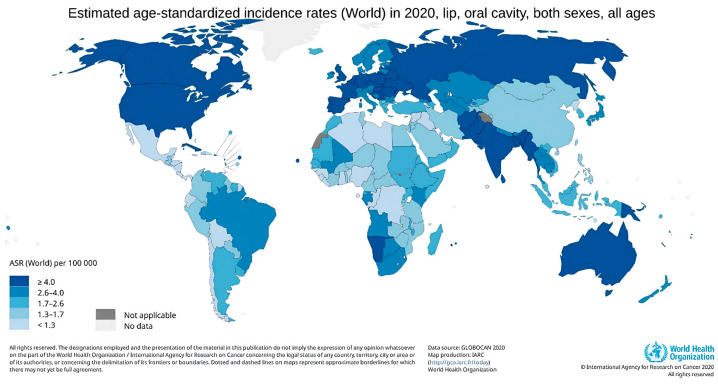
Global age-standardized incidence rates of lip and oral cavity cancer. Data source: GLOBOCAN 2020. Map production: IARC (http://gco.iarc.fr/today, accessed on 10 September 2024). World Health Organization.

**Figure 3 cancers-16-03156-f003:**
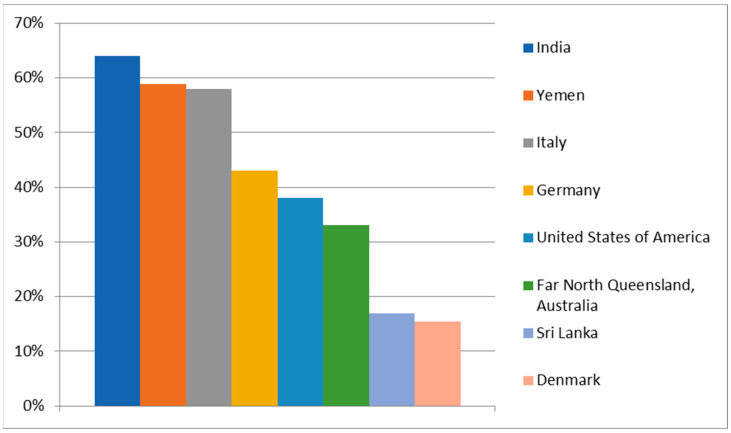
Percentage of People’s Awareness of Alcohol Consumption as a Risk Factor for Oral Cancer [46,47,48,49,50,51,52,53].

**Figure 4 cancers-16-03156-f004:**
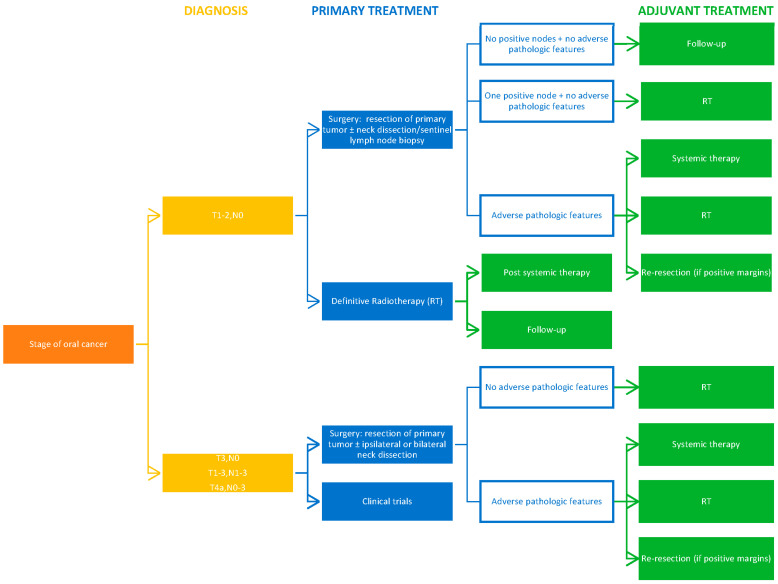
A decision tree summarising the current NCCN Clinical Practice Guidelines on treatment of cancer of the oral cavity.

**Table 1 cancers-16-03156-t001:** WHO classification of tumors of the oral cavity (2022) [19].

Localization	Histogenesis	Type
Oral cavity and mobile tongue tumors	Non-neoplastic lesions	Necrotizing sialometaplasia Multifocal epithelial hyperplasia Oral melanoacanthoma
	Epithelial tumors	Papillomas ○ Squamous papilloma Potentially malignant oral disorders and oral epithelial dysplasia ○ Oral epithelial dysplasia ○ Proliferative verrucous leukoplakia ○ Submucous fibrosis ○ HPV-associated oral epithelial dysplasia Squamous cell carcinomas ○ Oral squamous cell carcinoma ○ Verrucous carcinoma of the oral cavity and mobile tongue ○ Carcinoma cuniculatum
	Tumors of uncertain histogenesis	Congenital granular cell epulis Granular cell tumor Ectomesenchymal chondromyxoid tumor Melanotic neuroectodermal tumor of infancy
Oropharyngeal tumors	Benign lesions	Hamartomatous polyps
	Epithelial tumors	Squamous cell carcinoma, HPV-associated Squamous cell carcinoma, HPV-independent

**Table 2 cancers-16-03156-t002:** Examples of species inhabiting the oral cavity and their potential impact on carcinogenesis [82].

Bacteria	Role in Oral Cancer
*Fusobacterium nucleatum*	Secretes IL-1beta (key mediator of the inflammatory response)
*Neisseria*	Protective carotenoid biosynthesis and ALA metabolism pathways, reduces cancer risk
*Porphyromonas gingivalis*	Stimulates NF-κB/STAT3 signaling pathway by up-regulating pro-inflammatory cytokines Secretes PorKLMNP Core Complex (induction of chronic inflammation and interaction with host cells) Production of nucleoside diphosphate kinases (effects on cellular energy levels, regulation of proliferative processes, cell migration, and inhibition of apoptosis) Up-regulation of cyclins and p53 inhibition Induction of epithelial–mesenchymal transition through overexpression of β-catenin; chronic inflammation induction through IL-6, IL-8, TNF-α, and TGF-β1 expression Production of reactive oxygen species, butyrate, and acetaldehyde
*Prevotella intermedia*	Production of lipopolysaccharides, peptidoglycans, and lipoteichoic acid Expression of IL-1, IL-6, IL-8, IL-17, IL-23, and TNF-α,TNF-γ Secretion of proteases (promoting extracellular matrix degradation, altering cell signaling, regulating inflammatory processes, modifying cell structure, and promoting angiogenesis) Production of hydrogen sulfide, methyl mercaptan, and acetaldehyde
*Pseudomonas aeruginosa*	Activation of NF-κB pathway to Induct inflammation DNA damaging
*Treponema denticola*	Dentilisin overexpression may increase tumor invasiveness
*Lactobacillus*	Produces an antimicrobial compound, phenylacetic acid Some produce lactate (produce hydrogen peroxide) Protects the epithelial barrier, reduces inflammation, and regulates the microenvironment Induces cancer cell apoptosis by downregulation of MAPK pathway

**Table 3 cancers-16-03156-t003:** Clinical symptoms that may indicate oral cancer.

Clinical Symptoms That May Indicate Oral Cancer
Non-healing, unresponsive to the induced treatment, localized modification of appearance [88,89]; Rapid growth of lesion in a short period of time [88,89,90]; Pain localized in oral area [88,89]; Bleeding from suspected lesion [88,89]; Tooth mobility [88,89,91].

**Table 4 cancers-16-03156-t004:** A table summarising the characteristics of each diagnostic method.

Diagnostic Method	Pros	Cons
Vital tissue staining	Non-invasive Affordable Can help in choosing the biopsy locations and determining the boundaries of the lesion	High probability of false-negative results Questionable specificity and sensitivity May not provide information about deeper tissue layers
Optical imaging	Non-invasive Good sensitivity Helpful in the early diagnosis of suspicious lesions that may be missed under incandescent light	High cost Requires expertise to interpret images accurately
Oral cytology	Non-invasive Can be done during a regular dental examination Positive result requires additional tests or procedures such us a biopsy	May not detect all types of oral diseases, especially those affecting deeper tissues Cells can sometimes be misinterpreted, leading to incorrect diagnoses.
Salivary biomarkers	Non-invasive Can indicate oral cancer Can be developed for rapid detection and monitoring of disease progression	Biomarker levels can fluctuate and may not always be reliable Not all diseases have identifiable salivary biomarkers Requires advanced technology
Artificial intelligence	Non-invasive Helpful in the analysis of salivary biomarkers Can predict the probability of oral cancer development Assists pathologists	Performance depends on the quality and quantity of data used for training AI might miss nuances that a trained clinician would notice
Colposcopy	Non-invasive New promising approach for detecting oral lesions	Involves a physical examination and may require biopsy Primarily used for cervical and vaginal examinations; still not suitable enough for other areas
Spectroscopy	Non-invasive Early cancer detection Real-time assessment of surgical margins	Requires expertise to interpret spectroscopic data accurately High cost May not always be able to analyze deeper tissues effectively

**Table 5 cancers-16-03156-t005:** A table summarizing the current research on oral cancer immunotherapy.

Type of Immunotherapy	Description
Immune checkpoint inhibitors	Blockage of PD-1, PD-L1, CTLA-4, IDO1, TIM-3, LAG-3, TIGIT, VISTA, and their combinations [164,169,170].
Targeted monoclocal antibodies	Monoclonal antibodies targeting the receptors responsible for the pathogenesis and progression of oral cancer, e.g., EGFR, HER2, HER3, IGFR, VEGF, and CD44 [171].
Gene therapy	OR7C1 gene expressed in oral cancer cells located within the tumor can be a stem cell target [172].

## Data Availability

The original contributions presented in the study are included in the article, further inquiries can be directed to the corresponding author.

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
