# Peer review of "Epidemiology, Diagnostics, and Therapy of Oral Cancer—Update Review"

_cancers, 2024, doi:10.3390/cancers16183156_

Round 1

Reviewer 1 Report (Previous Reviewer 1)

Comments and Suggestions for Authors

The current review on oral cancer treatments includes surgery, radiotherapy, chemotherapy, and newer therapies such as immunotherapy and photodynamic therapy. Emphasis has been placed on the methods for which these modalities have effectiveness and limitations, with a focus on areas of debate and ongoing research. Novel treatments under development, coupled with the refinement of existing protocols, require further investigation to optimize their benefits in patients. Some comments are listed below for authors attention 

1.     The references are generally well used, but most of the claims would be further supported if more information was added or certain things were explained. For example, stating that "annual medical examinations" result in a "90% chance of survival" is a huge statement; it would have been much more convincing if studies or data behind this argument were discussed.

2. More updated references are required for Sections 1 and 2, particularly in the description of recent advancements in diagnostic and treatment techniques. The references cover a huge number of years by the date of publication, and it would have been good to see more recent studies highlighted, especially when dealing with a field as fast-moving as oncology.

3. A small overview of the reasons why classification and epidemiology are critical to understanding oral cancer could provide a stronger foundation for the detailed discussion that follows.

4. One would then understand the diverging trends in oral cancer incidence from one country to another on the basis of healthcare policies, lifestyle changes, or other socio-economic factors.

5. The statement that "people's awareness of the problem is relatively low" about alcohol and cancer could be supported by more recent data or at least by different data targeting a different demographic or geographic area.

6. The depth of discussion about the topics was patchy. For instance, there is a disproportionate emphasis on alcohol compared with the HPV and diet section, which feels comparatively brief and lacks mechanistic depth.

7. The sentence "the specific role of HPV in oral cancer development is still unclear" within the portion on HPV may appear contradictory when, later on, a strong causal link is suggested. This would cause confusion regarding where the literature currently stands with respect to the role of HPV in oral cancer.

8. One finds the section on diagnostic techniques to be quite well expounded, but perhaps somewhat less integrated with the previous discussion about risk factors. Closing the intellectual gap between the appreciation of risk factors and the importance of early diagnosis might have made this section more cohesive. For instance, some commentary on how certain risk factors, such as alcohol or HPV, may even inform a choice of diagnostic methods would help in this respect. 9. The argument could have greater coherence by linking sections to various treatment modalities. For instance, in the section that discusses radiotherapy, it would have been beneficial to make a comparison with surgery so that one can understand why one or the other may be advised in particular cases. The section on novel treatments is separate from the rest of the discussion.

9. Finally, it might be beneficial to explicitly link these innovations back to the limitations of current treatments to provide some rationale for why these new approaches are needed.

10. Some sections, such as immunotherapy, include full references and discussion, while others, such as chemotherapy, are superficial. There needs to be equal depth and rigor in the discussion of each modality for cancer treatment.

11. While the table summarizing the immunotherapy strategies in this section provides data that would have been very useful to readers, perhaps other figures or tables exemplifying changes in outcome with the various modalities of treatment, or summary depictions of some more complicated information given within the text, are warranted.

Author Response

Reviewer 2 Report (Previous Reviewer 2)

Comments and Suggestions for Authors

The manuscript titled,” A recent insight of oral cancer - update review” has the following issues:

1.       Although the manuscript has improved significantly, authors fail to maintain consistence when representing numerals “with 389.485 new diagnoses and 188,230 deaths estimated in 2022”.  Decimals should be use all throughout the text.

2.       Add a pictorial representation of different disease condition.

3.       In Fig3, the targets of the molecules mentioned in the second column should be appropriately elaborated. Maybe a 3rd column can be used for that purpose. In another column also include all references that support these data.

4.       In Fig4, authors have included facts which should be supported through clinical references. So please add them appropriately.

5.       Since the authors do not have enough biological literature particularly highlighting the signaling in oral cancer, chemoresistance, tumorigenesis; so I believe it might be appropriate to change the title so that the word therapy or treatment can be included, eg: A brief update on recent developments in oral cancer therapy

Round 2

Reviewer 1 Report (Previous Reviewer 1)

Comments and Suggestions for Authors

No further comments !

Reviewer 2 Report (Previous Reviewer 2)

Comments and Suggestions for Authors

Authors have clarified my concerns

This manuscript is a resubmission of an earlier submission. The following is a list of the peer review reports and author responses from that submission.

Round 1

Reviewer 1 Report

Comments and Suggestions for Authors

This paper presents an overview of the current modalities for treating oral cancer according to the disease stage and characteristics. It defines the role of multidisciplinary teams, decision-making, and up-to-date discussions about new therapies, such as immunotherapy and novel technologies, to assess margins. Future directions are focused on advancing treatment efficacy through ongoing research and clinical trials. Although this will be an excellent reading, some comments below have been put forth for the attention of the authors

Sections 1+ 2

1. " The introduction of screening programs on a larger scale could be a good initiative..." Specify which populations or regions would benefit most from these programs and cite examples of successful screening programs for other diseases or regions.

2.  "with almost 380.000 new diagnoses and 178.000 deaths estimated in 2020..."  Update statistics to the most recent data available and ensure consistency in formatting numbers

3. "annual medical examinations and early diagnosis of malignant lesions give 90% chances of survival after treatment." Clarify the most effective types of treatments and discuss advancements in therapy that could improve survival rates.

4. Epidemiology section: There is a lack of clarity in the text. For example, it is confusing when it says, "The best measure of cancer incidence is the incidence rate." Furthermore, it would be beneficial to explain why the rates are higher in certain regions rather than just listing factors, such as betel nut chewing.

5. Risk Factors: The role of acetaldehyde is very technical and requires some simplification for understanding. The statistics regarding awareness of the alcohol-cancer risk of Americans do not go very much with the rest of the section.

6. The explanation of the role of HPV in oral cancer is not as detailed as that in oropharyngeal cancers. Further context about current research investigating HPV's role of HPV in oral cancer is warranted.

7.   Oral Hygiene: This section needs further explanation of the mechanisms by which bacterial products participate in cancer. The comment that brushing teeth reduces cancer risk by 6% for each additional daily brush seems exciting, but a bit out of place compared with other text in the section. 

8. Oral Cytology:  This section discusses the accuracy and reliability of oral cytology compared to other available methodologies.

9. More specific illustrations of the application of AI in oral cancer diagnostics may go a long way toward strengthening this section.

10. Colposcopy:  This is a small section and might be further elaborated to provide more information on the efficiency and feasibility of this test.

11.  Repetitive phrases like "novel therapies are continuously being researched" and "further research is required" can be consolidated to avoid redundancy.

12. The section on novel treatments describes several developing therapies, but perhaps might better have focused on their status as research investigations and their future potential impacts.

13. The surgical section regarding margin assessment and intraoperative technologies is detailed but could, in part, be more concise. Some of the ideas that sound repetitive, such as the techniques used to assess margins, could all be put together into just one complete paragraph for better flow. More clearly defined subheadings and perhaps a more structured format would help the flow of information through the piece.

14.  Add examples or case studies to illuminate points about effectiveness and multidisciplinary approaches to treatment. More in-depth explanations of the new treatments, perceived research status of each at the time, and potential future impacts.

Author Response

Dear Reviewer

With kin regards,

Authors

Reviewer 2 Report

Comments and Suggestions for Authors

The manuscript titled,” A recent insight of oral cancer - update review” is highlighting a very important area of cancer that needs urgent introspection but in the present format this version of manuscript has some serious concerns as follows:

1.       Authors have completely missed out on discussing the molecular pathways that are necessary for survival of oral cancer cells. A diagrammatic representation is required to elucidate it and mention the different signaling nodes eg. Autophagic protection during chemotherapy.

2.       Authors need to highlight the use of Areca nut, betel leaves from different perspectives in terms of global oral cancer burden. Annually India has 100, 000 new cases of oral cancer and contributes to one third of global oral cancer burden. Try to highlight different hot spot areas and their causes according to GLOBOCAN hotspots. Authors have cherry picked certain countries and mentioned their statisctics.

3.       Also mention how immunologically cold the oral cancer tumors can be. A figure will be good to support immunotherapy routes.

4.       In fig 3 add columns highlighting their pros and cons

5.       Replace Fig 1 with higher resolution version

6.       Introduction is written very superficially.

7.       Avoid repeating same sentences. Eg: “Cancers of lip and oral cavity are the 17th most common cancer in the world [1].” Has been repeated

8.       In line 41, check the numbers

9.       Classification is very poorly written.

Overall assessment: This is DEFINITELY not a critical review. It appears more like a flash card, with certain topic details which have been cherry picked and all grounds and details are missing.

Author Response

Dear Reviewer

With kind regards,

Authors

Reviewer 3 Report

Comments and Suggestions for Authors

The authors aimed to review the current knowledge of oral cancer, including the classification, epidemiology and recent advances in diagnostic and treatment techniques.

The review has a sound structure. However, the authors failed to provide the recent knowledge they claimed. For example, the authors used the 2017 WHO classification of OSCC. However, the WHO updated this classification in 2022 and released it in print in 2023. Likewise, the authors cited the 2007 paper describing the WHO classification of oral potentially malignant disorders, published in 2021, with several significant changes to the old classification.

This observation is noticed in all review aspects, and I don't think it provides recent updates. Another critical point is that the authors didn't focus on using evidence from the recent systematic reviews and meta-analyses to support their statement since there is almost a systematic review on each of their review subheadings. For example, about adjunctive tools, here is a recent review "Adjunctive aids for the detection of oral squamous cell carcinoma and oral potentially malignant disorders: A systematic review of systematic reviews. Jpn Dent Sci Rev. 2024;60:53-72. doi:10.1016/j.jdsr.2023.12.004".

The review includes several incorrect and irrelevant details that have no association with oral cancer. For example, what is the purpose of discussing hairy leukoplakia when it has no link to oral cancer?

Including "sticky saliva or its absence" is unacceptable as a sign of oral cancer. If that is true, all patients with dry mouth could have oral cancer. The scientific community will be absurd by this statement. 

Comments on the Quality of English Language

Some grammatical errors need fixing. 

Author Response

(The authors gave the same response as above.)
